# Revisiting the Role of the Leucine Plug/Valve in the Human ABCG2 Multidrug Transporter

**DOI:** 10.3390/ijms26094010

**Published:** 2025-04-24

**Authors:** Orsolya Mózner, Kata Sára Szabó, Anikó Bodnár, Csenge Koppány, László Homolya, György Várady, Tamás Hegedűs, Balázs Sarkadi, Ágnes Telbisz

**Affiliations:** 1Institute of Molecular Life Sciences, HUN-REN Research Centre for Natural Sciences, Magyar Tudosok krt. 2, 1117 Budapest, Hungary; mozner.orsolya@ttk.hu (O.M.); katasara98@gmail.com (K.S.S.); anibodnar24@gmail.com (A.B.); csenge.koppany@gmail.com (C.K.); homolya.laszlo@ttk.hu (L.H.); varady.gyorgy@ttk.hu (G.V.); 2Doctoral School, Semmelweis University, 1085 Budapest, Hungary; 3Department of Biophysics and Radiation Biology, Semmelweis University, Tuzolto u. 37–47, 1094 Budapest, Hungary; tamas@hegelab.org; 4HUN-REN TKI-SE Biophysical Virology Research Group, 1052 Budapest, Hungary; 5Salus Kft, Than Károly utca 20, 1119 Budapest, Hungary

**Keywords:** ABCG2, BCRP, MXR, multidrug transporter, leu plug/valve variants

## Abstract

In the human ABCG2 (ATP Binding Casette transporter G2/BCRP/MXR) multidrug transporter, a so-called “leucin plug/valve” (a.a. L554/L555) has been suggested to facilitate substrate exit and the coupling of drug transport to ATPase activity. In this work, we analyzed the effects of selected variants in this region by expressing these variants, both in mammalian and Sf9 insect cells. We found that, in mammalian cells, the L554A, L554F, L555F, and a combination of L554F/L555F variants of ABCG2 were functional, were processed to the plasma membrane, and exhibited substrate transport activity similar to the wild-type ABCG2, while the L555A and L554A/L555A mutants were poorly expressed and processed in mammalian cells. In Sf9 cells, all the variants were expressed at similar levels; still, the L555A and L554A/L555A variants lost all transport-related functions, while the L554F and L555F variants had reduced dye transport and altered substrate-stimulated ATPase activity. In molecular dynamics simulations, the mutant variants exhibited highly rearranged contacts in the central transmembrane helices; thus, alterations in folding, trafficking, and function can be expected to occur. Our current studies reinforce the importance of L554/L555 in ABCG2 folding and function, while they do not support the specific role of this region in selective substrate handling and show a general reduction in the coupling of drug transport to ATPase activity in the mutant versions.

## 1. Introduction

ABC (ATP Binding Casette) transporters are present in a wide range of organisms and transport a wide variety of substances coupled to ATP binding and hydrolysis. These latter processes are provided by conserved intracellular ABC domains, whereas substrate specificity is determined by the transmembrane regions. Human ABC multidrug transporters reside in important tissue barriers and work together in defense against various chemically unrelated compounds by extruding toxic endo- and xenobiotics [1,2,3]. Among these, the ABCG2 multidrug transporter is present in the plasma membrane of several cells and tissue barriers, including the blood–brain barrier, kidneys, intestines, and stem cells [4,5,6,7]. The importance of ABCG2 has been shown in the absorption, distribution, metabolism, excretion (ADME), and toxicity of various drugs, and has a role in the multidrug resistance of cancer cells [8,9]. Besides causing drug resistance, ABCG2 is also important in uric acid excretion, and mutations in the *ABCG2* gene contribute to a genetic susceptibility to gout [10,11]. Being a multidrug transporter, the substrates of ABCG2 vary in structure and size; thus, understanding the molecular transport mechanism of ABCG2 should help with drug design and therapeutic decisions. Investigating the interactions of a new drug under development with ABCG2 is recommended by both the FDA and EMA [12,13].

The current ABCG2 structural models are based on high-resolution atomic structures and knowledge about similar ABCG transporters, including ABCG2 [14,15,16,17], ABCG5/G8 [18], and ABCG1 [19]). However, the exact drug-binding site(s) and the molecular pathway for exporting various drugs have still not been fully uncovered, although molecular dynamic simulations along with mutational studies should help to understand the biochemical details of drug transport. Recent cryo-EM structures and studies of designed mutants at structurally important locations have suggested that the opening and closing of the molecular extrusion pathway is controlled by a “leucine valve/plug” in the near-extracellular part of transmembrane domains (positions 554L and 555L). ABCG2 functions as a homodimer, and this leucine valve is suggested to be formed by residues of both monomers [3,14,15,20].

However, several details of these earlier studies were controversial. A report indicated that the L555A mutant could not be expressed, while the L554A variant was expressed and showed significantly higher baseline ATPase activity, with reduced stimulation by E1S (estrone-1-sulfate), although E1S transport was accelerated [15]. In another study, the L554 and L555 residues were substituted by alanine, isoleucine, and cysteine, and functionality was tested in HEK293 cells [20]. The authors concluded that all L555 mutants showed strongly reduced expression; still, the mitoxantrone transport capability of the L555A mutant was retained and this variant had increased ATPase activity. This conclusion is partially in contrast with our previous study in Sf9 insect cells [21], which reported that the L555A variant could be expressed normally, but the functionality was compromised. In this report, we also showed that the mutation of the L558 residue to alanine, localized near the 554–555 residues, did not change the transport activity, while the drug stimulation of ATPase activity was decreased.

In recent studies, the homologous positions to the proposed “leucine valve” of ABCG2 have also been shown to have an important function in other ABCG-type proteins. The human ABCG1, ABCG4, ABCG5, and ABCG8 transporters are involved in sterol transport, the homologous sites are often phenylalanines instead of leucines (F570 and F571 in ABCG1, F576 and M577 in ABCG8, and L549 and F548 in the ABCG5 of the ABCG5/G8 heterodimer), and sterol binding is localized near these residues [19,22]. In the case of the ABCG36 plant protein, when the corresponding L704 residue was mutated to phenylalanine, a significant change in the transported substrate specificity was observed [23].

In the present work, we aimed to revisit the analysis of this region, and our hypothesis was that mutating the “valve” leucines to alanine or phenylalanine in ABCG2 may alter the function and substrate recognition of this transporter. The variants examined in our experiments were the following: L554A, L555A, L554A-L555A (DMA) double mutant to alanine, L554F, L555F, and L554F-L555F (DMF) double mutant to phenylalanine. In HEK293 cells, both the transient and stable expression of these mutants was achieved. We measured the cell surface expression of ABCG2 through the binding of the 5D3 antibody, which recognizes an extracellular epitope of ABCG2, while the total ABCG2 protein expression was estimated by Western blotting. EGFP (Green Fluorescent Protein) was expressed from the constructs using an IRES2 (Internal Ribosome Entry Site 2) element; thus, ABCG2 was not tagged, while the cellular expression levels could be followed by the simultaneous eGFP expression levels. ABCG2-dependent Hoechst33342 transport experiments were performed in live cells by using flow cytometry. In parallel experiments, ABCG2 variants were also expressed in Sf9 insect cells, which are known to properly express even those membrane protein variants that are hard to express in mammalian cells. Both ABCG2-ATPase and substrate transport measurements were performed in isolated membranes of the ABCG2-expressing Sf9 cells.

## 2. Results

### 2.1. Expression and Function of ABCG2 Variants in a Human Cell Line, HEK293

The ABCG2 variants examined in HEK293 cells were L554F, L555F, and the L554F-L555F (DMF) double phenylalanine mutant, as well as the alanine mutants L554A and L555A and the L554A-L555A (DMA) double alanine mutant. During some of the experiments, in addition to the WT-ABCG2, the well-characterized, naturally occurring Q141K variant was also used as a control (the Q141K-ABCG2 was shown to have reduced cell surface expression due to a folding deficiency, but retained function [24,25]). The ABCG2 variants were either investigated after transient expression or stably inserted in the genome via the Sleeping Beauty transposon method. ABCG2 expression in the applied construct was driven by a CAG promoter and the transcribed mRNA contained an untagged ABCG2 and an eGFP code, separated by an IRES2 element [26]. This enabled the simultaneous expression of ABCG2 and eGFP, and the transfection efficiency could be monitored based on GFP fluorescence in the cells. Stable cell lines could also be generated by sorting GFP-positive cells after the genomic insertion of the sequence by the Sleeping Beauty transposon system.

According to the Western blot results shown in Figure 1A,B, the phenylalanine variants were expressed at similar levels to the WT in the transiently transfected cells, while the alanine mutants showed decreased levels of ABCG2, especially if L555 was mutated. When compared to the WT-ABCG2, L554A showed a slightly lower expression level and L555A resulted in a large decrease in ABCG2 protein expression, while the double alanine mutant L554A-L555A (DMA) showed almost negligible protein expression.

To investigate the presence and amount of ABCG2 in the plasma membrane, cell surface expression measurements were performed in live cells via flow cytometry after labeling the cells with the 5D3 antibody, which recognizes the extracellular epitope of ABCG2. Levels of the ABCG2 variants on the cell surface were found to be similar to the WT-ABCG2 for the L554F, L555F, DMF, and L554A variants, but the L555A and DMA cell surface levels (58% and 38%, respectively) were even less than that for the Q141K (77%) used as a reference for reduced surface level (Figure 1C). These data were similar in both experimental setups, where the ABCG2 variants were either transiently or stably expressed in HEK cells.

The functionality of the ABCG2 variants was examined by using the transported fluorescent substrates mitoxantrone and Hoechst33342, and the uptake was measured in HEK293 cells via flow cytometry. In the following experiments, using cells transiently expressing ABCG2, the transport activity was measured 48 h after transfection. The cells were gated for GFP expression to include only ABCG2-positive cells, and the dye uptake was measured in the absence (active transport) and in the presence (fully inhibiting ABCG2 function) of Ko143. The MAF (multidrug resistance activity factor) values, corresponding to the relative transport activity, were calculated from these measurements (for details, see Section 4).

As documented in Figure 2A,B, the ABCG2-L555A-expressing cells showed a significantly decreased Hoechst33342 and mitoxantrone transport activity, and the L554A-L555A double mutant ABCG2 showed practically no mitoxantrone transport and a strongly decreased Hoechst transport activity. In these experiments, we found no significantly altered transport activity for L554A, L554F, L555F, or the L554F-L555F double mutant (from the phenylalanine variants, only the double mutant (DMF) is shown in Figure 2A).

In the following experiments, we wanted to explore if the diminished substrate transport rate of the ABCG2 L555A and DMA mutants was a functional defect or the consequence of a lower cell surface presence for these variants. For these studies, we generated stable ABCG2-expressing HEK cell lines by the Sleeping Beauty transposon method (see Section 4.1), and then the cells were sorted based on equal levels of cell surface 5D3 anti-ABCG2 antibody staining by flow cytometry. The transposon-based integration of the expression constructs may cause various protein expression levels, but this method enabled the selection of stable cell lines expressing similar amounts of the examined ABCG2 variants at the cell surface. As shown in Figure 3A, by using this method, we could select cells with similar, relatively high levels of ABCG2 expression on the cell surface, even from the L555A- and DMA-expressing HEK cell lines. In this way, we could measure the ABCG2 transport function at nearly equal cell surface expression levels for all the ABCG2 variants. In these experiments, we used relatively higher Hoechst33342 concentrations (up to 10 μM) to explore potential functional differences with increased substrate loading.

As shown in Figure 3B, although there was a similar plasma membrane expression level in the sorted ABCG2-HEK cells, we still observed a significantly impaired function for the L555A and L554A-L555A ABCG2 variants. These experiments indicate that these variants have a major folding problem, which does not only reduce their trafficking to the cell membrane, but also directly affects their substrate transport function.

### 2.2. Expression and Function of Human ABCG2 Variants Expressed in Sf9 Insect Cells

It has been shown that a baculovirus-based insect cell protein expression system can be successfully used for membrane protein expression, even in the case of significant folding and processing problems of these proteins in mammalian cells [27,28]. Therefore, we used Sf9 cells here to express the Leu valve ABCG2 variants, and membrane vesicles were prepared from the baculovirus-infected Sf9 cells for the functional characterization of these variants.

In Sf9 membranes, ABCG2 variant expression was followed by a Western blot analysis. As shown in Figure 4A, we found a high level and a comparable expression of all the examined ABCG2 variants, similar to that of the wild-type ABCG2. It should be mentioned that ABCG2 from Sf9 expression runs a bit lower in SDS-PAGE gels due to the shorter glycosylation produced by insect cells. Batch-to-batch variations may be significant when using this expression system; therefore, two independent batches for the wild-type ABCG2 expression level are presented in Figure 4A.

The transport activity of the ABCG2 variants was measured in inverted membrane vesicles by following the ATP-dependent Lucifer yellow substrate uptake. Lucifer yellow (LY) is a relatively hydrophilic substrate transported by WT-ABCG2, and some mutant variants (e.g., R482G ABCG2) are unable to transport LY, although they exhibit Hoechst dye and mitoxantrone transport [29]. The Sf9 cell membrane has a low cholesterol level, which results in a low transport activity for ABCG2, and it has been shown that this transport activity can be greatly increased by loading cholesterol into the Sf9 membrane vesicles [30]. In the present experiments, cholesterol–cyclodextrin was applied for achieving this high ABCG2 transport activity (see Section 4), while without added cholesterol, all the ABCG2 variants showed very low or undetectable transport activity [30].

As shown in Figure 4B, most of the ABCG2 variants examined (L554F, L555F, DMF, and L554A) showed active LY transport activity except the L555A and DMA mutant variants. Still, as compared to the wild-type ABCG2 variant, with a similar membrane expression level, the L554F, L555F, DMF, and L554A variants had only about one third of the LY transport activity. In all cases, the specific inhibitor Ko143 abolished the ABCG2-mediated LY uptake into the membrane vesicles.

ABC multidrug transporters show vanadate-sensitive membrane ATPase activity connected to their transport function; therefore, measuring this ATPase activity is a widely used method to follow drug interactions or functional changes in ABC transporter variants. As shown in Figure 4C, most of the ABCG2 mutants examined here showed significant vanadate-sensitive and Ko143-inhibited basal ATPase activity, except L555A (which showed only slight activity) and the DMA mutant, which had no such specific ATPase activity. Interestingly, this basal activity was relatively high (almost twice as high as that of the wild-type ABCG2) in the case of the L554F and DMF variants.

Quercetin is a transported drug for ABCG2, and it significantly stimulates the ATPase activity of the wild-type transporter [21,30]. As documented here (Figure 4C), while the activity of the wild-type ABCG2 was about 2.5 times higher in the presence of quercetin, this stimulation was much less in the case of all L554 and L555 variants (although the L554F and DMF variants had relatively high basal ATPase activity).

A similar phenomenon was observed when using prazosin, another ABCG2 substrate, to stimulate the ATPase activity. In contrast to the major (2.5×) stimulation of the wild-type ABCG2 ATPase, in the mutant variants, prazosin only negligibly stimulated this ATPase (see below, Table 1, and Appendix A).

To explore the potential differences in the quercetin dependence of this ATPase activation, we measured this effect at a wide range of quercetin concentrations (Figure 4D). We found that the lack of significant quercetin activation for the L554F, L555F, and L554F-L555F (DMF) variants was not due to a major shift in the concentration dependence of the ATPase activation, but to a general quercetin insensitivity of this activity. Again, the L555A and L554A-L555A (DMA) ABCG2 variants had low ATPase activity, and no significant activation was observed over the whole range of quercetin concentrations.

Since membrane cholesterol has a significant stimulatory effect on the function of the ABCG2 transporter, we also studied the cholesterol dependence of the ATPase activity of the variants. As shown in Table 1 and in Appendix A, in the case of wild-type ABCG2, cholesterol loading of the Sf9 cell membranes had little effect on the basal ATPase activity, while the quercetin or prazosin stimulation of the ATPase activities was significantly increased upon cholesterol loading. In the case of the L554F, L555F, L554F-L555F (DMF), and L554A variants, cholesterol loading only slightly increased both the basal and the drug-stimulated ATPase activities; thus, the influence of cholesterol was much less pronounced. In Appendix A, we show the effect of a few other known modulators and substrates on the ATPase activity of ABCG2 mutants. These auxiliary results, not discussed here in detail, are in accordance with the main experiments.

### 2.3. MD Simulations for the Leu-Valve Mutations in ABCG2

Since the mutations in the Leu valve likely exert their effects on the orientation of the central four helices (TM2 and TM5 from both protomers), we performed molecular dynamics simulations with an inward-open conformation of ABCG2 (PDBID: 6HIJ, [31]). The trajectories from these equilibrium simulations were stable and did not reveal differences in the dynamic fluctuations of the WT, L554A-555A, or L554F-L555F constructs (Appendix A). Therefore, we explored the contacts between the central helices, including TM2 and TM5 from one half and TM2 and TM5 from the other half of the transporter, since static differences can be expected to develop along the simulations.

Only very few contacts with low frequencies were observed between TM5 (chain B) and any part of TM2 (chain B or chain A) (Figure 5A,B). Similarly, low contacts were detected between the TM5 (chain A) and TM2 helices (Appendix A). Interestingly, the Ala mutations at positions 554 and 555 resulted in a greatly increased number of non-native contacts between TM5 (chain B) and TM2 (chain B) from the same protomer (Figure 5C).

The contacts at the extracellular ends between TM5 (chain B) and TM2 (chain A) from the opposite protomer were also increased in the L554A-L555A ABCG2 (e.g., 555A/N425 and 555A/G428) (Figure 5D). The majority of the non-native contacts between TM5 (chain B) and TM2 (chain B) did not develop in the L554F-L555F ABCG2, except those between residues T538/V442, T538/F445, and V534/E446 (Figure 5E). The non-native contacts present between the opposite TM2 and TM5 in L554A-L555A were also present in the L554F-L555F ABCG2 (Figure 5F). All these phenomena were observed for the TM5 (chain A) interactions from the other protomer (Appendix A). It is important to note that differences can be observed in the dynamics and interactions of the two halves of the protein, despite the homomeric nature of the ABCG2 dimer, that may have arisen from the not-completely-symmetrical initial structure.

## 3. Discussion

A conserved feature of ABC membrane transporter proteins is substrate transport coupled to the activity the cytoplasmic ATPase domain. However, the site and mode of substrate binding and transport through the dimeric protein structure show great variability. For multidrug transporters of clinical relevance, understanding the similarities and differences in the broad substrate specificity is important for drug development [3]. These proteins are important factors in drug resistance, which is a conspicuous problem in the treatment of cancer and other chronic diseases. In addition, in the case of multiple drug therapies, unwanted side effects can be more severe if the drugs used interact with and inhibit multidrug transporters.

The key features of the molecular pathways of substrate translocation and the identification of inhibitor-binding sites are still under investigation and debate [17,32,33]. The proposals are based on structural models verified by cryo-EM studies and experimental analyses of designed key point mutations in the protein [15,17,34,35,36]. Previously, it was suggested that the transport pathway in ABCG2 is gated by leucines L554 and L555, which constrict and block the translocation pathway in the upper transmembrane region and may play a role in the switch between the inward and outward conformations [14,15,31,37].

However, in other ABCG-type protein structures, leucines in the corresponding positions are often replaced by phenylalanine residues [19,38,39], and these amino acids are important for the substrate specificity and transport activity of the protein. The loops that form a hydrophobic gate in the corresponding region of ABCG1, a cholesterol and lipid transporter, contain phenylalanines [19,40]. By comparing the cryo-AM-based structure of ABCG1 and that of ABCG2, it appears that the wider cavity of ABCG2 allows interactions with larger drug substrates, whereas the cavity of ABCG1 can only accommodate a flat sterol molecule. In both proteins, the hydrophobic gate is in the same position, but in ABCG1, F570 and F571 are present instead of the L554 and L555 of ABCG2 [19]. A mutational study confirmed that the substitution of F571 for alanine in ABCG1 results in preserved basal ATPase activity, but altered sterol and rhodamine123 substrate interactions [22].

In the ABCG5/G8 heterodimer protein, a cholesterol and sitosterol transporter, the corresponding residues (F576 and M577 in ABCG8, and L549 in ABCG5) participate in a structure that may serve for the transport of flat sterol molecules. However, in this protein, the absence of a bulky amino acid at the fourth position suggests an altered translocation pathway [19,38]. A computational model suggests that conserved phenylalanines near this position may play a role in ligand capture [41]. The significance of the leucine valve was further emphasized by a mutational study of a plant ABCG protein. Certain mutations of the L704 amino acid of the Arabidopsis ABCG36/PDR8/PEN3 broadened the substrate specificity of this transporter without uncoupling the ATPase and transport activity [23].

We found that the L-to-F substitution of valve leucines (L554F, L555F, and L554F-L555F) within the ABCG2 protein, when expressed in HEK293 cells, did not result in alterations in the total cellular or membrane expression levels, in comparison to the WT-ABCG2 protein. Furthermore, the transport functions of these variants, when measuring Hoechst33342 or mitoxantrone extrusion, were similar to that of the wild-type ABCG2. However, in detailed functional studies performed in an Sf9 insect cell expression system and using isolated membrane vesicles, significantly decreased Lucifer yellow transport (although still cholesterol-dependent) was observed for these ABCG2 variants. Furthermore, the L554F and L554F-L555F variants exhibited elevated basal ATPase activities and diminished levels of quercetin and prazosin stimulation, in comparison to the WT protein.

These results indicate that the 554–555 phenylalanine variants of ABCG2 preserve the basic folding and trafficking features, and only subtle changes in substrate recognition may occur in the variants. Such changes, however, are hardly recognizable in a cellular system. The results also demonstrate that the substrate translocation pathway remains functional despite these mutations, suggesting an impaired coupling of the transport cycle to the ATPase activity of the cytosolic domain.

In a series of parallel experiments, the leucine residues at positions 544 and 555 were mutated to alanine in the HEK293 cell expression system. It was observed that the cell surface expression and the transport function of the L554A variant remained unaffected, while the L555A and, most notably, the L554A-L555A ABCG2 variants experienced significant folding and trafficking problems. Furthermore, experiments measuring the transport activity of these ABCG2 variants in HEK cells, sorted to have a similar, stable transporter expression, revealed impaired Hoechst dye transport by the L555A and especially the L554A-L555A variants. These results suggest that, in addition to the folding/trafficking problems, these L-to-A mutations have a direct effect on the transport function of ABCG2.

In the Sf9 insect cell expression system, the L554A variant demonstrated functionality, yet exhibited a diminished level of substrate stimulation of the ATPase activity and a reduced Lucifer yellow transport capacity. The L555A and L554A-L555A mutant variants were also properly expressed in the insect cells, while the specific substrate transport function and the related ATPase activity of these latter variants were practically lost. This is in contrast to a previous report, which showed increased uncoupled ATPase activity with somewhat impaired substrate transport, probably due to expression problems [20]. This loss of function indicates severe structural changes in the L554A-L555A ABCG2 variant, suggesting a more pronounced structural problem than presented in an earlier report.

Our molecular dynamics simulations support the above conclusions. While the L555F mutation exerted only minor effects on intramolecular contacts, our simulations highlighted large changes in residue contacts among the central helices in the L554A-L555A protein, when compared to the wild-type and L554F-L555F double mutant variants. The Ala mutations at positions 554 and 555 resulted in a highly increased number of non-native contacts between the TM5 and TM2 helices.

Our previous molecular dynamics studies [38,39] indicated that the orientation of the TM helices may play a role in substrate recognition, and the cholesterol regulation of ABCG2 function may also involve the TM helix orientation. In the R482G-ABCG2 variant, TH3 (transmembrane helix 3) moved closer to TH4 and drifted away from TH1 [38], altering the shape of a potential lateral substrate-binding pocket or interaction site. The presence of cholesterol in the membrane bilayer, which is also required for ABCG2 function, promoted the closure of the intracellular ends of ABCG2 [39]. Similarly, in our current study, the central TM helices exhibited an altered conformation upon mutations in the “Leu valve” region, affecting the available conformational space of the transporter and consequently altering the transport properties.

The present findings underscore the significance of the “valve” or “plug” region residues in the ABCG2 protein. Furthermore, these results suggest moderate changes in transport function and the ATPase coupling of drug transport when leucines 544 and 555 are exchanged for phenylalanines. The L554F mutation was found to have a more significant role in this respect. The results indicate that these ABCG2 residues do not play a key role in substrate recognition, which can be mostly attributed to the central cavity, as evidenced by recent structural models [17,42]. Instead, our results demonstrate a significant impact on the protein expression, folding, and trafficking of the L554A and L555A variants, particularly the double mutant L554-L555A, in mammalian cells. This has been shown to result in a substantial reduction in the amount of protein that reaches the plasma membrane. Indeed, the combined expression and functional measurements suggest that, in this variant, even if expressed at high levels, both substrate transport and ATPase activity are strongly impaired.

It is hoped that the present studies, performed to facilitate a more profound understanding of the molecular transport mechanism of ABCG2, will make a contribution to the development of a more sophisticated molecular model. The development of such a model will, in turn, allow for improved predictions of drug–transporter interactions and the development of clinically applicable inhibitors.

## 4. Materials and Methods

### 4.1. Mammalian Expression System—Cell Culturing and Cell Line Generation

HEK293H cells were grown in a DMEM/high-glucose/GlutaMAX medium (Gibco, Billings, MT, USA, cat. 10569010) completed with 10% FBS (Gibco, cat. 1640071) and 1% penicillin–streptomycin (Gibco, cat. 15070063) at 37 °C (5% CO_2_). The transfection of HEK293 cells was carried out with Lipofectamine 2000 (Invitrogen, Waltham, MA, USA, cat. 11668019) in the Opti-MEM medium (Gibco, Billings, MT, USA, cat. 31985070), according to the manufacturer’s protocol. The ABCG2-expressing stable cell lines were generated using the Sleeping Beauty transposon–transposase system. We used a 1:9 SB100X transposase coding plasmid: pT4 transposon plasmid ratio for transfection [43,44,45]. Three days after transfection, the eGFP-positive cells were sorted using BD FACS Aria II and seeded on 6-well plates. Another sorting was performed after two more weeks of culturing.

### 4.2. ABCG2 Transport Function Measurements in Live Cells by Flow Cytometry

The Hoechst 33342 (Hst, Thermo Fisher, Waltham, MA, USA, cat. H1399) and mitoxantrone (Sigma Aldrich, St. Louis, MO, USA, cat. M2305000) uptake was determined on stable HEK293 cell lines. Cells were trypsinized and then preincubated for 5 min at 37 °C with or without 1 µM of the selective ABCG2 inhibitor, Ko143 (MedChemExpress, Monmouth Junction, NJ, USA, cat. HY-10010). Hoechst33342 dye (1 µM in all cases except stable cell lines, where 10 µM was used in cells sorted for equal high ABCG2 surface expression) or mitoxantrone (1 µM) was added to the cells and incubated at 37 °C for 20 min. The drug accumulation fluorescence (Hoechst or mitoxantrone) of the eGFP-positive cells was measured using a violet laser (405 nm) and VL1 detector for Hoechst, and a red laser (637 nm) and RL1 detector for mitoxantrone accumulation on an Attune NxT Cytometer. The MAF (multidrug resistance activity factor) values were calculated from the median averaged fluorescence results in the following way: MAF = (F (inh) − F (no inh))/F (inh), where F (inh) is the median of the Hoechst or mitoxantrone fluorescence of the cells with the inhibitor, while F (no inh) is the fluorescence of the cells without the inhibitor.

### 4.3. ABCG2 Cell Surface Expression Level Measurements by Flow Cytometry

Antibody labeling was performed on trypsinized cells with the ABCG2-specific 5D3 mouse monoclonal antibody (gift of Bryan Sorrentino, Division of Experimental Hematology, Department of Hematology/Oncology, St. Jude Children’s Research Hospital). Ko143 is an ABCG2 inhibitor that has an impact on ABCG2 conformation, thus helping 5D3 antibody recognition. 1 µM of Ko143 was added to the samples before the antibody labeling. Alexa Fluor 647-labeled IgG2b (Thermo Fisher, Waltham, MA, USA, cat. A-21242) was used as a secondary antibody. Measurements were carried out by the Attune NxT Cytometer after gating for live, EGFP-positive cells.

### 4.4. Western Blot

The total protein from the HEK293 cells or Sf9 membranes was extracted by the addition of a TE sample buffer (0.1 M TRIS-PO4, 4% SDS, 4 mM Na-EDTA, 40% glycerol, 0.04% bromophenol blue, and 0.04% β-mercaptoethanol; materials from Sigma-Aldrich). Equal amounts of the protein samples, as determined by the Lowry method, were loaded on 7.5% (Sf9 membranes) or 10% (HEK) acrylamide gels. PVDF blots were probed with the following primary antibodies: anti-ABCG2 (BXP-21, Abcam, Cambridge, UK, cat. ab3380) and anti-β-actin (Sigma Aldrich, St. Louis, MO, USA, cat. A1978). Goat anti-mouse IgG (H+L) HRP conjugate (Abcam, Cambridge, UK, cat. ab97023) secondary antibodies were used to visualize and quantify the results. Detection was performed with the Clarity Western ECL Substrate (BioRad, Hercules, CA, USA, cat. 1705060) and the BioRad ChemiDoc Imaging System. A densitometry analysis was performed by the ImageLab 6.0.1 (BioRad, Hercules, CA, USA) and ImageJ software v1.42q.

### 4.5. Generation of ABCG2-Expressing Sf9 Insect Cells and Membrane Preparation

Sf9 insect cells were grown in suspension at 27 °C in a TNM-FH medium (Sigma-Aldrich, cat. T3285) supplemented with 5% FBS. ABCG2 variants expressing baculovirus vectors were constructed (pACUW vector, expression from p10 promoter as in our previous papers) and combined with the Flashback ULTRA Kit (Oxford Expression Technologies Ltd., Oxford, UK) according to the manufacturer’s protocol. A membrane fraction was prepared from infected cells after 72 h by mechanical disruption and a differential centrifugation method, as described earlier [28], and stored at −80 °C. The cholesterol content of the Sf9 membranes was increased by loading the membranes during the membrane preparation procedure, as described in [30]. The total protein content was measured by the Lowry method and ABCG2 expression was detected on a Western blot, as described above.

### 4.6. ATPase Activity Measurement in ABCG2-Sf9 Membrane Vesicles

The ATPase activity of WT-ABCG2 and the mutant variants was measured by the colorimetric detection of inorganic phosphate liberation in microplates, as described previously [28]. The protein concentrations (5 μg/well) were normalized according to Western blots for ABCG2; thus, equal amounts of the ABCG2 variants were used in the measurements. The cholesterol loading of the membranes was achieved by the addition of 0.6 mM RAMEβ-cholesterol (Cyclolab Ltd., Budapest, Hungary) before the measurements, and the membranes were incubated for 10 min on ice, according to [30]. The measurement of the ATPase activity was started with the addition of 3.1 mM MgATP, the samples were incubated at 37 °C, and the reaction was terminated after 25 min with the P-reagent. Photometric measurements were carried out in a Victor Multilabel plate reader (PerkinElmer, Waltham, MA, USA) at 660 nm. The ABC-transporter-specific activity was determined as vanadate-sensitive ATPase activity (1 mM of Na-vanadate was applied). The ABCG2-specific ATPase activity was verified by adding the ABCG2-specific inhibitor Ko143 to the reaction. The other investigated drug compounds were applied in DMSO (the final DMSO concentration was below 1% and this amount of DMSO had no effect on the results). The compounds (Merck, Darmstadt, Germany) and concentrations are given in the main text.

### 4.7. Lucifer Yellow Transport Assay in ABCG2-Sf9 Membrane Vesicles

Sf9 membrane vesicles (30 μg protein/sample) were incubated at 37 °C for 10 min (without or with 4 mM of Mg-ATP) in a 50 μL volume, in the presence of the ABCG2-transporter-specific fluorescent substrate Lucifer yellow (LY; 5 μM). The specificity and quality were controlled by the reference inhibitor Ko143 (1 μM). The Mg-ATP (3 mM)-dependent uptake was measured [30]. After incubation, the samples were rapidly filtered and washed on filter plates (MSFBN6B10, Millipore, Burlington, MA, USA). The accumulated substrates in the vesicles were resolved from the filter by 100 μL of 10% sodium dodecyl sulfate and centrifuged into another plate. A 100 μL volume of fluorescence stabilizer was added to the samples (DMSO). The fluorescence of the samples was measured by a plate reader (Victor X3 PerkinElmer, Waltham, MA, USA) at appropriate wavelengths (405/535 nm) for LY. The ABCG2 protein-related transport was calculated by subtracting the passive uptake, measured without Mg-ATP.

### 4.8. Statistical Analysis

All the experiments included at least 3 biological replicates with at least 2 technical parallel measurements. GraphPad Prism8 was used for the data analysis and visualization. The Western blot results, ABCG2 cell surface expression results, and flow-cytometry-based transport results were analyzed by one-way ANOVA. Dunnett’s multiple comparisons test (95% confidence interval) was performed when comparing the results to the WT-ABCG2 results. Columns marked with stars show significant differences compared to the WT-ABCG2 (adjusted *p*-value < 0.05).

### 4.9. Molecular Dynamics

The apo, inward-open ABCG2 structural model determined by cryo-EM (PDB ID: 6HIJ) was subjected to loop modelling [31]. We favored this conformation, since the transmembrane helices of the ATP-bound structures are packed tightly; thus, their movements are sterically limited and do not exhibit changes upon mutations on the time scale of molecular dynamics simulations. Shorter missing regions (a.a. 49–58, 302–327, and 355–368) were built using the standard loop modelling method of Modeller 9.23 [46].

The input files for all the steps (energy minimization, equilibration, and production run) were generated by the CHARMM-GUI web interface [47,48] by submitting the full-length WT structure, oriented according to the OPM database [48]. The mutations were also introduced at CHARMM-GUI. The membrane bilayer was asymmetric, containing 35:25:25:15 cholesterol/POPC/PLPC/SSM and 35:18:17:17:8:5 cholesterol/POPC/PLPC/POPE/POPS/DMPI25 (POPC: 1-palmitoyl-2-oleoylphosphatidylcholine; PLPC: 1-palmitoyl-2-linoleoylphosphatidylcholine; SSM: stearoyl-sphingomyelin; POPE: 1-palmitoyl-2-oleoylphosphatidylethanolamine; POPS: 1-palmitoyl-2-oleoylphosphatidylserine; DMPI25: dimyristoylphosphatidylinositol-4,5-bisphosphate). The KCl concentration was set to 150 mM in TIP3 water, the grid information for PME (particle-mesh Ewald) electrostatics was generated automatically, and a temperature of 310 K was set. The structures were energy-minimized using the steepest descent integrator (maximum number to integrate: 50,000 or converged when force is <1000 kJ/mol/nm). From the energy-minimized structures, parallel equilibrium simulations (3 for each system) were forked, followed by production runs for 1 µs. The Nosé–Hoover thermostat and Parrinello–Rahman barostat were applied. Electrostatic interactions were calculated using the fast smooth PME algorithm [49], and the LINCS algorithm was used to constrain bonds [50]. A constant particle number, pressure, and temperature ensembles with a time step of 2 fs were used. The simulations were performed with a CHARMM36m force field using GROMACS 2020 [51,52].

The root mean squared deviation (RMSD) and root mean squared fluctuation (RMSF) values were calculated using GROMACS tools. Contact maps were generated using the MDAnalysis 2.8.0 Python package [53]. The RMSF and contact maps were derived from a trajectory obtained by merging the equilibrated portions (last 250 ns) of three parallel simulations. Two residues were considered to be in contact if their Cα atoms were within 7 Å of each other. These contacts were recorded for each residue pair across all frames of the merged trajectory. The contact frequency was then calculated by dividing the number of observed contacts by the total number of frames (n = 75,000). A contact frequency of 1 indicates that the residues were in contact in every frame, 0.5 means they were in contact in half of the frames, and 0 indicates no contact throughout the trajectory. The plots were generated using the Matplotib Python package https://matplotlib.org/.

## Figures and Tables

**Figure 1 ijms-26-04010-f001:**
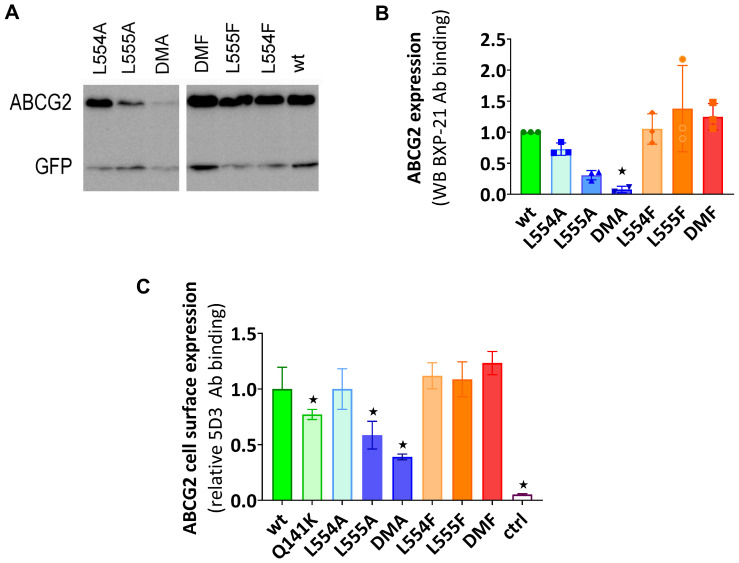
(**A**,**B**) The expression of ABCG2 variants in HEK cells, determined by Western blotting using the ABCG2-specific BXP-21 antibody and simultaneously developing the anti-GFP signal (mean ± SD, n = 3). The ctrl sample was parental HEK transfected by an empty vector. The original blot pictures are in the Appendix A. (**C**) The cell surface expression of the variants, measured by 5D3 binding in cells with transient ABCG2 expression. The 5D3 antibody binds to an extracellular epitope, and the antibody binding was measured in living cells by flow cytometry (mean ± SD, n = 3). Columns marked with a star showed a significant difference (*p* < 0.05) compared to the WT-ABCG2 sample in the ANOVA analysis, corrected with Dunnett’s multiple comparisons test.

**Figure 2 ijms-26-04010-f002:**
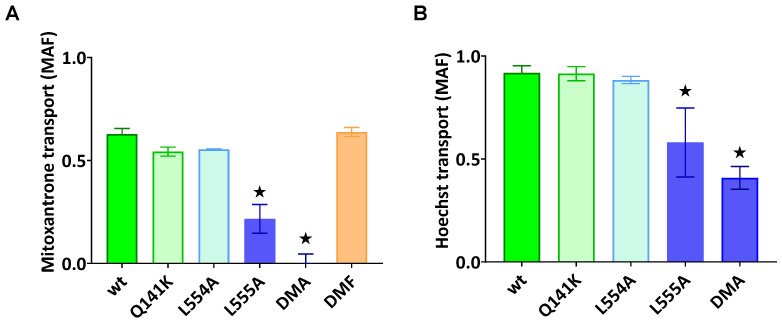
ABCG2-mediated mitoxantrone (**A**) and Hoechst33342 (**B**) transport in HEK cells transiently expressing the ABCG2 variants. MAF (multidrug resistance activity factor) values are given for the characterization of the transport (definition is in Section 4). Mean ± SD, n = 2 biological parallels each with 2 technical parallels. The columns marked with a star showed significant (*p* < 0.05) differences compared to the WT-ABCG2 sample in ANOVA, using Dunnett’s multiple comparisons test.

**Figure 3 ijms-26-04010-f003:**
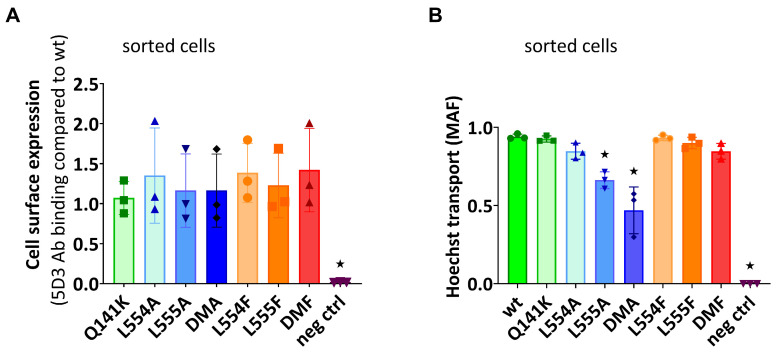
(**A**) The cell surface level of the ABCG2 protein variants and (**B**) the Hoechst transport function in cells sorted for equal cell surface ABCG2 expression. (**A**) The cell surface expression of the ABCG2 variants after sorting stable cell lines for similar ABCG2 cell surface expression by 5D3 labeling. (**B**) The Hoechst dye extrusion function (MAF values) in stable ABCG2-HEK293 cell lines expressing similar amounts of the ABCG2 variants on the cell surface. Columns marked with a star showed significant differences (*p* < 0.05) compared to the WT-ABCG2 sample using ANOVA after Dunnett’s multiple comparisons test (mean ± SD, n = 3).

**Figure 4 ijms-26-04010-f004:**
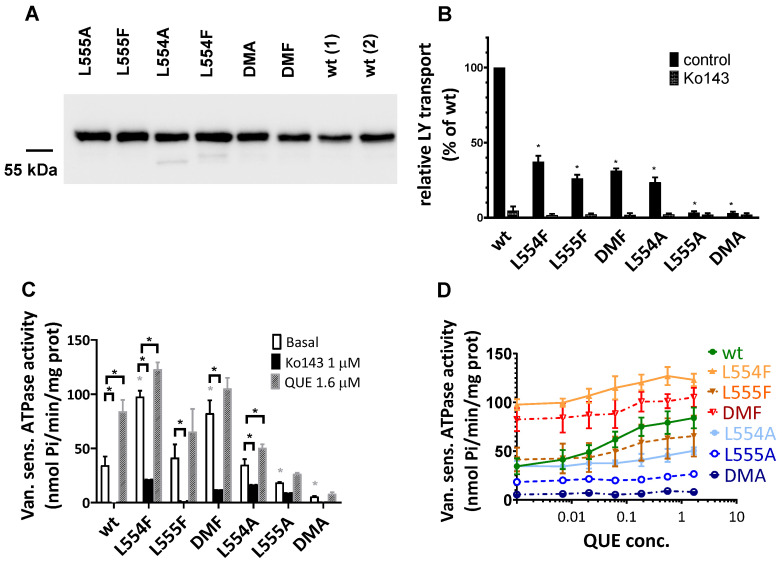
Expression level and function of the ABCG2 variants in insect cell membrane vesicles. (**A**) The expression of the ABCG2 variants measured in Sf9 membrane vesicles prepared from recombinant baculovirus-infected insect cells. The same amounts of membrane proteins were applied in each well and BXP-21 antibody staining was applied for the detection of the ABCG2 protein. The original blot pictures are in the Appendix A. (**B**) The LY transport activity of ABCG2 variants in membrane vesicles. n = 4; 3 parallels in each experiment; mean ± SD; significant differences, indicated by *p* < 0.01, are labelled by a black * (Student’s *t*-test). (**C**,**D**) Vanadate-sensitive (labelled as Van. sens.) ATPase activity of the ABCG2 variants in Sf9 membrane vesicles. The substrate-transport-connected ATPase activity was measured by adding quercetin as a probe substrate and Ko143 as a specific inhibitor. n = 2 (3 parallels in each experiment); mean ± SD; significant differences, indicated by *p* < 0.01, are labelled by a grey * for the comparison of basal values between WT and a given mutant and a black * for the comparison of treatment values (Student’s *t*-test).

**Figure 5 ijms-26-04010-f005:**
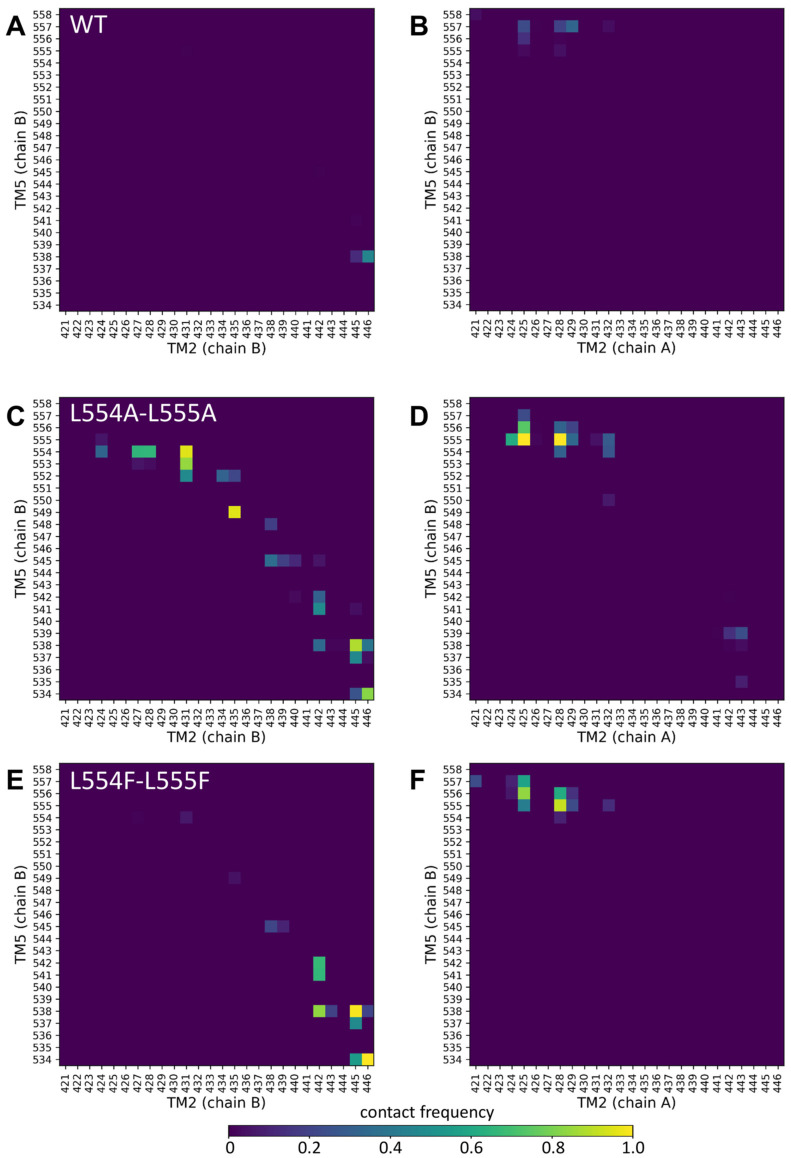
Altered contacts of the TM2 and TM5 helices. Contacts between TM2/TM5 from the same protomer (chain B/chain B) (**A**,**C**,**E**) and TM2/TM5 from the opposite protomers (chain A/chain B) (**B**,**D**,**F**) were calculated between Cα atoms at a 7 Å cutoff. Other combinations of these central TM helices are shown in Appendix A.

**Table 1 ijms-26-04010-t001:** Effect of cholesterol and substrate drugs on the ATPase activity of ABCG2 variants, measured in Sf9 membrane vesicles. Related graphs are shown in the Appendix A.

Relative ATPase Activities—Drug-Stimulated/Basal Activities (Average ± Standard Error) ^1^					
	Chol+/ No Chol	+QUE/ No Chol	+QUE/ Chol+	+PRAZ/ No Chol	+PRAZ/ Chol+
WT	0.89 ± 0.12	1.50 ± 0.16	2.44 ± 0.18 **	0.6 ± 0.12	1.50 ± 0.22 **
L554F	1.08 ± 0.04	1.08 ± 0.1	1.26 ± 0.04	0.73 ± 0.05	1.02 ± 0.07
L555F	0.95 ± 0.17	1.38 ± 0.18	1.58 ± 0.3 *	0.88 ± 0.11	1.03 ± 0.19
L554F-L555F	1.03 ± 0.09	1.11 ± 0.05	1.28 ± 0.07	0.77 ± 0.05	1.03 ± 0.06
L554A	1.16 ± 0.06	0.36 ± 0.04	1.45 ± 0.04 *	0.87 ± 0.01	0.89 ± 0.05

^1^ Data show the ratio of the drug-stimulated (1.6 µM quercetin (QUE) or 20 µM prazosin (PRAZ)) to “basal” activity. Data also indicate relative ATPase activities in the original low-cholesterol Sf9 membranes (no Chol), compared to those upon the addition of 2 mg/mL cholesterol-CD (Chol+). n = 3, 3 parallels in each experiment, average ± SEM, * *p* < 0.01, ** *p* < 0.001, Student *t*-test.

## Data Availability

The article and Appendix A contain all the data.

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
