# Peer review of "Revisiting the Role of the Leucine Plug/Valve in the Human ABCG2 Multidrug Transporter"

_ijms, 2025, doi:10.3390/ijms26094010_

Round 1
Reviewer 1 Report
Comments and Suggestions for Authors
This article conducted a comprehensive investigation of leucine plug/valve on ABCG2 protein. A minor revision is suggested for this article. Several suggestions are attached below:
- The sequence of figure should be rechecked and rearranged. For example, the appearance of Supplementary Figure 3A is earlier in the introduction?
- Some data were missing. For the WB, please provided the original blot of three repeats.
- Some “significance mark” were missing, such as Fig. 1B, Fig 4C, and several figures in the supplementary data. Please go through all of the figures and do the correction. Also, the format of “significance mark” should be uniform throughout the article.
- Some typing errors exist, such as line 135, there are two stop signs, line 264-265, too much spacing.
- In the human and Sf9, the author used different fluorescent substrates, therefore it’s difficult to compare the results. LY efflux (transport) experiment should also be performed in the human cell lines.
- Figure 1C, the final bar “ctrl” should have an explanation of what it is.
- In the discussion, most of the information was repetitive results. The author should review literature and previous studies and include some discussion of the clinical importance of this “valve” or “plug” region residues in the ABCG2 protein. For example, how often if this variant appears in the human samples? Will it lead to a change in the physiological function or pathological condition? Will it influence the clinical treatment in some diseases?
Author Response
Thank you for the careful reading of our paper. We accepted the critical remarks and corrected the manuscript in the following points.
- The sequence of figure should be rechecked and rearranged. For example, the appearance of Supplementary Figure 3A is earlier in the introduction?
The figure sequence was checked and corrected. (Reference to figures were removed from introduction and methods)
- Some data were missing. For the WB, please provided the original blot of three repeats.
We provided additional WB pictures in a supplementary file: Original western images.
- Some “significance mark” were missing, such as Fig. 1B, Fig 4C, and several figures in the supplementary data. Please go through all of the figures and do the correction. Also, the format of “significance mark” should be uniform throughout the article.
We accept the reviewer's point and labeled significance in Figures 1B, 4C and Supplementary Figures.
- Some typing errors exist, such as line 135, there are two stop signs, line 264-265, too much spacing.
Corrected.
- In the human and Sf9, the author used different fluorescent substrates, therefore it’s difficult to compare the results. LY efflux (transport) experiment should also be performed in the human cell lines.
We agree with the reviewer that different substrates may be transported at different rates, so we did not intend to compare them. Our aim was to analyse as many substrates as possible, as it is known that mutations can differentially affect the binding of different substrates, as observed for R482 mutants (line 206-209, ref 18)
We have to emphasize that it is not possible to measure LY efflux in human cells because LY is a hydrophilic substance that cannot normally enter cells (perhaps electroporation could be used to load cells, but it has not been investigated whether ABCG2 can measurably expel LY from cells, and also the presence of other transporters may be problematic in this situation).
Perhaps we did not emphasise enough that Sf9 membrane vesicles are inverted vesicles, where LY can easily be added to the buffer and ABCG2 transports the substrate to the inside of the vesicle. We have included this information in the new version " Transport activity of the ABCG2 variants was measured in inverted membrane vesicle…".(line 205) It should be mentioned that mitoxantrone or Hoechts uptake cannot be measured in membrane vesicles because passive diffusion is high in membrane vesicles.
- Figure 1C, the final bar “ctrl” should have an explanation of what it is.
We inserted the description of the ctrl in Figure 1 text: „ctrl sample was parental HEK transfected by an empty vector”
- In the discussion, most of the information was repetitive results. The author should review literature and previous studies and include some discussion of the clinical importance of this “valve” or “plug” region residues in the ABCG2 protein. For example, how often if this variant appears in the human samples? Will it lead to a change in the physiological function or pathological condition? Will it influence the clinical treatment in some diseases?
We accepted the point and supplemented the abstract with a more detailed discussion of related work. We also added some paragraphs that put our work into a broader context of potential applicability. However, this research is in the field of molecular biology, which helps to understand protein properties and may have relevance to drug development. The designed mutations are not clinically relevant forms of this protein, so we do not intend to discuss clinical issues in this paper. Our group has published other studies of SNP variants and we have discussed clinical aspects in these papers.
New parts of the discussion are the following:
lines 323-330, 339-350 (more detailed discussion of literature), 426-430
Reviewer 2 Report
Comments and Suggestions for Authors
Mozner et al. present a manuscript dealing with a special domain, the leucine valve, in the ABCG2 drug transporter. By mutating two leucines to alanine or phenylalanine and expression of the mutants in mammalian and insect cells they show that phenylalanine mutations have little or no impact on folding, trafficking and transport activity of ABCG2, whereas the alanine mutations have a much stronger effect. Several aspects are tested, surface transport, ATPas activity, transport of several substrates. The study is carefully designed, the data look convincing, results are well described. Novel molecular details on ABCG2 are provided. I recommend the paper for publication, the only small comment I have is that maybe the molecular dynamics simulations could be a bit more explained in detail. What is a contact frequency of 1, for example? Is this a stable interaction? And in the legend of suppl. Fig. 3, what are RMSD plots and what is RMSF? I appreciate showing the original images. There is only a small mistake, the Western Blot corresponds to Fig. 1A, not 2A. The figure should be provided as a suppl. Fig.
Author Response
Thank you for the careful reading of our paper and its acceptance. We accepted the critical remarks and corrected the manuscript in the following points.
- What is a contact frequency of 1, for example? Is this a stable interaction? And in the legend of suppl. Fig. 3, what are RMSD plots and what is RMSF? I appreciate showing the original images.
We are sorry for the sparse information on these MD-related text.
Now we inserted explanation of the contact frequency into the Method section as:
„Two residues were considered to be in contact if their Cα atoms were within 7 Å of each other. These contacts were recorded for each residue pair across all frames of the merged trajectory. Contact frequency was then calculated by dividing the number of observed contacts by the total number of frames (n = 75,000). A contact frequency of 1 indicates that the residues were in contact in every frame, 0.5 means they were in con-tact in half of the frames, and 0 indicates no contact throughout the trajectory.”
We supplemented the legend of Suppl Figure 3 with the detailed meaning of RMSD and RMSF as:
“(B) The ABCG2 proteins were stable in MD simulations. RMSD (Root Mean Square Deviation) plots, calculated relative to the initial structure, show values not exceeding 5–6 Å, indicating the stability of a protein of this size during the simulations (n = 3 for each system). (C) No notable differences in dynamics were observed among the ABCG2 variants. RMSF (Root Mean Square Fluctuation) values were calculated using conformations from the final 250 ns of the simulations. These values represent the positional fluctuations of each Cα atom around its average position. GROMACS tools also convert RMSF values into pseudo B-factors, allowing numerical comparison with B-factors derived from X-ray crystallography. These pseudo B-factors are embedded in the PDB files, allowing the structures to be visualized using colored wire representations, where regions of higher flexibility appear as thicker wires in warmer colors.”
- There is only a small mistake, the Western Blot corresponds to Fig. 1A, not 2A. The figure should be provided as a suppl. Fig.
Corrected and more blots are added to a supplementary file: Original western images.